# Using Enriched Category Theory to Construct the Nearest Neighbour Classification Algorithm

## Abstract

This paper is the first to construct and motivate a Machine Learning algorithm solely with Enriched Category Theory, supplementing evidence that Category Theory can provide valuable insights into the construction and explainability of Machine Learning algorithms. It is shown that a series of reasonable assumptions about a dataset lead to the construction of the Nearest Neighbours Algorithm. This construction is produced as an extension of the original dataset using profunctors in the category of Lawvere metric spaces, leading to a definition of an Enriched Nearest Neighbours Algorithm, which, consequently, also produces an enriched form of the Voronoi diagram. Further investigation of the generalisations this construction induces demonstrates how the $k$ Nearest Neighbours Algorithm may also be produced. Moreover, how the new construction allows metrics on the classification labels to inform the outputs of the Enriched Nearest Neighbour Algorithm: Enabling soft classification boundaries and dependent classifications. This paper is intended to be accessible without any knowledge of Category Theory.

## 1 Introduction

As Machine Learning (ML) becomes more popular, the use of black-box approaches is beginning to hinder the progression of the field. During engineering and development, the better one understands a model, the easier it is to improve its performance, diagnose faults, and provide guarantees for its behaviour. Unfortunately, necessary to the development of many algorithms, there are design decisions motivated by intuition or trial and error. Part of the difficulty in understanding these algorithms comes from a lack of clarity in how they interact with the data they are provided. How does the encoding of input data affect the information an algorithm understands? To approach this question, this paper seeks to investigate the use of Enriched Category Theory ($ECT$) for the design of Machine Learning algorithms. To provide evidence that this approach has potential, it is demonstrated that basic assumptions about a dataset can lead to the natural construction of a pre-existing algorithm, which is popular for its predictable and robust behaviour: the Nearest Neighbours Algorithm (NNA).

The argument for using Enriched Category Theory in such a theory proceeds as follows. The process of learning requires the ability to make comparisons. This may be comparisons between entries of a training dataset to identify patterns, between training examples and new cases for the sake of inference, and between different models of the same dataset. Enriched Category Theory provides a general framework for defining and studying comparisons between objects. It demonstrates that the entirety of the information associated with an object can be encoded in its comparisons to other objects. Using Enriched Category Theory, the structure of data can be encoded explicitly in their mutual comparisons rather than implicitly, as is familiar with many ML algorithms. The benefit of this approach would be that the design and mechanism of ML algorithms become more transparent. The assumptions about datasets can be made more explicit. Moreover, the learning process can be interpreted in its natural form as reasoning about the comparison of observations.

## 2 Background

To the knowledge of the authors, the construction of the Nearest Neighbours Algorithm demonstrated in this paper is one of the first examples of a machine learning algorithm motivated and constructed solely with Enriched Category Theory. There is one other example of an entirely categorical construction of an ML algorithm, where previous work (Shiebler, 2022) shows that the single linkage clustering algorithm can be found as a Kan-extension of a dataset of points. However, it is suggested that the steps shown for the derivation of the *NNA* draw a tighter parallel between the intuition of how the dataset is represented, and the derived algorithm.

There are also examples of algorithms whose structures have been encoded in the language of category theory, such as Graph Neural Networks (Dudzik & Veličković, 2022). But they represent the structure of how the algorithm computes information, and not necessarily the selection of the optimal model or representation of the input dataset. In contrast, the *NNA* construction draws a direct line from the representation of the data to the selection of the optimal classification.

Understanding the Enriched Category Theory construction of the Nearest Neighbours algorithm requires an understanding of Lawvere metric spaces as Cost Enriched Categories, as well as a working knowledge of the Nearest Neighbours Algorithm. It is beyond the scope of this paper to provide a complete introduction to Enriched Category Theory [1], but thankfully many of its complexities can be avoided by focusing on the specific case of Lawvere metric spaces. The following section provides the necessary components, as well as a brief overview of the Nearest Neighbours Algorithm.

### 2.1 Nearest Neighbours Algorithm

The Nearest Neighbours Algorithm (Fix & Hodges, 1989) extends the classification of a dataset of points in a metric space to the entire metric space. Consider a dataset of $n$ pairs $(x_1, y_1), ..., (x_n, y_n)$. The targets of the dataset, $y_i$, are elements of a set of class labels $Y$. The features of the dataset, $x_i$, represent points in a metric space $X$. This allows the distance between any two points to be measured, following the traditional metric space axioms.

- $d(a, a) = 0$
- $a \neq b \Leftrightarrow d(a, b) > 0$            *Positivity*
- $d(a, b) = d(b, a)$            *Symmetry*
- $d(a, b) + d(b, c) \geq d(a, c)$            *Triangle Inequality*

To a point of the metric space not in the dataset, the Nearest Neighbours Algorithm assigns a class if a closest point in the dataset has that class. An example of the classification regions produced can be seen in Fig 1 which shows the *NNA* classification of a two class dataset of points sampled from two Gaussian distributions.

To express this as a relation we can represent the dataset with two functions. The indexes of the dataset can be expressed as the set of integers from 1 to n, $N = \{a \in \mathbf{Z} \mid 1 \leq a \leq n\}$. The features of the dataset can be encoded with the function $F : N \to X$ such that $Fi = x_i$. The targets of the dataset can be expressed similarly with a function $T : N \to Y$, such that $Ti = y_i$. Given a point $x \in X$ and a class $y \in Y$, the relation should return true if the closest data-point to $x$ has the class $y$. [2]

$$NNA(y, x) = \exists i \in N \ [ \quad Ti = y \quad and \quad d(Fi, x) = \inf_{i' \in N} d(Fi', x) \quad ]$$

This relation can be presented in an alternate form that will be useful later, but it requires that the indexes are partitioned based on their classes. We define the partition as follows. $NT(y) = \{i \in N \mid Ti = y\}$. This

---

[1] A basic introduction can be found in "Seven Sketches in Compositionality" (Fong & Spivak, 2018) while a more technical overview occurs in "Basic Concepts of Enriched Category Theory" (Kelly, 2005).

[2] inf in the following expression represents the infimum or least upper bound of a set of values. For finite cases it can be replaced with minimum.

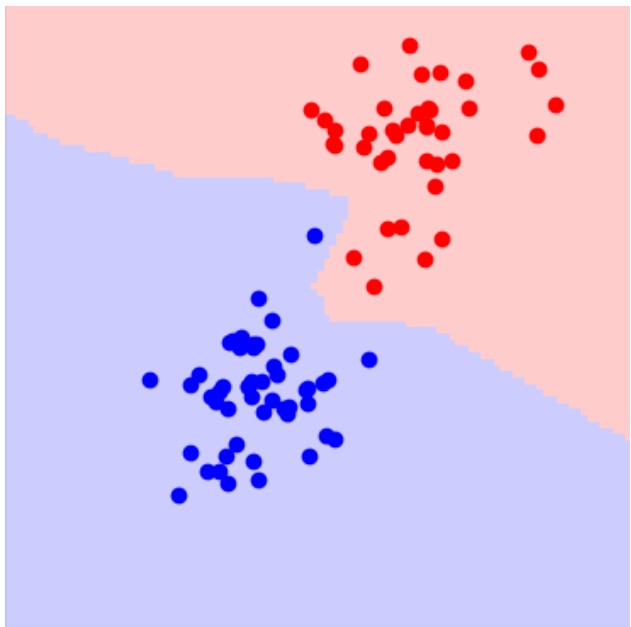

Figure 1: An example of the classification regions produced by the nearest neighbour algorithm from data points sampled from two Gaussian distributions, representing the distributions of the two classes.

allows the relation to be presented as:

$$NNA(y, x) \quad \Leftrightarrow \quad \inf_{i \in N} d(Fi, x) \quad = \inf_{i \in NT(y)} d(Fi, x)$$

## 2.2 Lawvere Metric Spaces

As mentioned in the introduction, Enriched Category Theory provides a method of encoding structure through a rigorous language for talking about comparisons. In some sense, an Enriched Category is a collection of objects which can be compared. Given a category $C$, two objects $x \in C$ and $y \in C$ can be compared with the notation $C(x, y)$. This is referred to as the hom-object of $x$ and $y$. This hom-object exists in its own category called the base of enrichment. To make the comparisons meaningful, $ECT$ requires that the base of enrichment have some way of combining hom-objects, called a monoidal product, and some juxtaposition of these two hom-objects to a third. An example of how this structure works can be seen in order relations. Consider a category called Fruits, which is a collection of fruits ordered by price. The hom-object $Fruits(Apple, Orange)$ would test to see if Apples were cheaper than Oranges. In this instance this comparison could also be written as $Apples \leq Oranges$. The outcome of this comparison is either true or false so the base of enrichment would be a category containing an object representing true and an object representing false. This base of enrichment can be called Bool for Boolean.

A sensible logical deduction to make with such a category would be to say that if we know fruit $A$ is cheaper than fruit $B$, and fruit $B$ is cheaper than fruit $C$, then $A$ must be cheaper than $C$. Notionally, this can be written as:

$$(A \leq B) \ and \ (B \leq C) \implies (A \leq C)$$

This process of logical inference gives the general motivating structure of an enriched category. In this instance, each comparison of the ordered set returns a value in Bool. The monoidal product of Bool is the logical "and", allowing its objects to be combined. The hom-object of Bool is logical implication.

$$Bool(x, y) = (x \Rightarrow y)$$

Because Bool can be described with a hom-object which takes values in itself, it is described as self enriched. When Bool is used as the base of enrichment, the general structure of the enriched category becomes the structure of a pre-order relation.

A Lawvere metric space is an enriched category whose base of enrichment is chosen so that the categories operate like metric spaces, allowing the enriched category to measure the distances between its objects. The base of enrichment for Lawvere metric spaces is called the Cost category. Because it represents measurements of distance, its objects are the non-negative real numbers extended with infinity[3]. Given a Cost enriched category $X$, and two objects $x$ and $y$ of $X$, the hom object $X(x, y)$ can be interpreted as the distance between $x$ and $y$. The monoidal product of Cost is addition. Cost can be interpreted as both a Bool enriched category, and a self enriched category. As a Bool enriched category, it is a preorder, with the ordering of its objects given by the standard order relation on the real numbers.

$$Cost(x, y) = (x \geq y)$$

Looking at the previous example, we can replace the *and* operation of Bool with addition, and the implication with $\geq$ to recover the following expression for Cost categories.

$$X(x, y) + X(y, z) \geq X(x, z)$$

This requirement of Cost categories is the triangle inequality, stating that taking a detour to a third object cannot be quicker than travelling directly between two objects. By choosing the Cost category as the base of enrichment, *ECT* naturally recovers some, but not all of the the metric space axioms (As detailed in section 2.1). This makes Lawvere metric spaces pseudo-metric spaces. In Lawvere metric spaces, one retains the triangle inequality, and the requirement that the distance from an object to itself is zero ($d(a, a) = 0$), but the metric spaces are not required to be symmetric ($d(a, b) = d(b, a)$) and two different objects can be zero distance apart. This can be a controversial choice, but there are several arguments for this being a desirable outcome. For example, in many cases an intuitive notion of distance is not symmetric, e.g. its easier to go down stairs than up them. One might also say that distance is a measure of similarity not identity, and the idea of two different objects being zero distance apart is sensible when considering systems at a certain level of coarseness. In either case, if one wishes to operate with traditional metric spaces, they are all also Lawvere metric spaces, and the necessary axioms can be asserted as convenient.

By sensibly considering how we wish to compare objects in our enriched categories, choosing objects and a monoidal product in the base of enrichment, we have recovered the structure of a metric space. Though the Lawvere metric space is one of the simpler examples of an enriched category, it starts to reveal the power of such a theory to construct complex structures for the representation of data.

## 2.3 Functors and Profunctors

An Enriched Category may be thought of as representing a particular datatype, with the structure of that datatype being represented by the hom-objects of the category. In order to interact with this information, there are many ways of comparing categories to each other. Between categories with the same base of enrichment, there are two constructions which are relevant for this work: Functors and Profunctors.

In set theory, a mapping from one set to another is called a function. In *ECT*, there is a similar concept called a functor. Functors between enriched categories are structure preserving maps. In the case of Cost-enriched categories (Lawvere metric spaces), this reduces to the statement that functors are distance non-increasing functions. Given a functor $F : X \to Y$, from X to Y, this can be expressed as the statement that for any two objects $a, b \in X$.

$$X(a, b) \geq Y(Fa, Fb)$$

As well as Functors between categories being the *ECT* version of functions between sets, there is also an *ECT* version of relations between categories. A set relation $R$ between two sets $X$ and $Y$ is often described

---

[3]The objects of Cost being $\{x \in \mathbb{R} \mid x \geq 0\} \cup \{\infty\}$. The monoidal product is addition, with addition by infinity defined as $x + \infty = \infty$

as a subset of the Cartesian products of $X$ and $Y$, i.e. $R \subseteq X \times Y$. However, this relation can also be thought of as a function which returns true if the relation is true, and false if the relation is false: $R : Y \times X \to \{False, True\}$. In $ECT$, this notion is extended to a functor from the tensor product of two categories to the base of enrichment.

$$R : Y^{op} \otimes X \to Cost$$

Such a construction is called a profunctor. For notation, a profunctor $R : Y^{op} \otimes X \to Cost$, can be written as $R : X \nrightarrow Y$. The tensor product of two categories $Y^{op} \otimes X$ contains objects which are pairs of objects in $X$ and $Y$ similar to how the Cartesian product of sets contains pairs of elements of sets. The notation $Y^{op}$ is used to refer to a category with the objects of $Y$, but whose hom objects are reversed.

$$Y(a, b) = Y^{op}(b, a)$$

With two set relations $R : X \nrightarrow Y$ and $S : Y \nrightarrow Z$, a composite relation can be produced of the form $S \circ R : A \nrightarrow C$. The composition of two relations $R$ and $S$ is true for two inputs $x$ and $z$, if there exists an element $y$ in $Y$ such that $R(y, x)$ is true, and $S(z, y)$ is true. The logic of relation composition is described by the following equation.

$$(S \circ R)(z, x) := \exists y \in Y[R(y, x) \text{ and } S(z, y)]$$

Similar to relations, profunctors can also be composed. Given Cost enriched profunctors $R : X \nrightarrow Y$ and $S : Y \nrightarrow Z$, the output of their composition bares a striking resemblance to the formula for relation composition.

$$(S \circ R)(z, x) := \inf_{y \in Y} (R(y, x) + S(z, y))$$

The similarity between relation composition and profunctor composition is more than just cosmetic. It also emulates how Cost enriched categories treat logical propositions. In the Boolean logic setting, the "and" operation outputs true only when both of its inputs are true, and false otherwise. In Cost enriched categories, a distance of zero can be interpreted as true, and a distance greater than zero is false. With this interpretation, the sum of two values $a$ and $b$, where both are non-negative, can only be zero if both $a$ and $b$ are zero. From the perspective of Cost category logic, $a + b$ is the logical "and" operation. Furthermore, within this version of logic the infimum operation is the Cost version of the existential quantifier. When X is finite, The statement $\inf_{x \in X} Fx = 0$ means there exists a value $x$ such that $Fx$ is zero. In the infinite case, it suggests that there exists a value $Fx$ which is arbitrarily close to zero. Applying this logic to the definition of profunctors, it can be seen that profunctors produce truth values from pairs of objects, if the output of zero is interpreted as true, and the output of non-zero is interpreted as false. Such an interpretation can be represented by the functor $(0 = x) : Cost \to Bool$. The following table highlights the comparable notions between standard Set theory and Lawvere metric spaces, though this comparison extends to all enriched category theory.

| Logical Concept | Set Theory | Lawvere metric spaces |
|---|---|---|
| Truth Values | Bool | Cost |
| Conjunction | $\wedge$ | $+$ |
| Mappings | Functions | Functors |
| Binary Predicate | Relations | Profunctors |
| Existential Quantifier | $\exists$ | inf |
| Universal Quantifier | $\forall$ | sup |

With knowledge of Functors, Profunctors and their composition there is a final piece of information necessary for the construction of the Nearest Neighbours Algorithm. Continuing with the intuition from functions and relations of sets, it can be observed that functions are a special kind of relation, known as a functional relation. A function $F : N \to X$ is said to produce an element $Fi$ when given an element $i \in N$, but this behaviour can be represented directly as a relation $F_* : N \nrightarrow X$ which evaluates to a truth value under the

condition $F_*(x, i) \Leftrightarrow (x = Fi)$. In fact, there is also a second relation of the opposite direction $F^* : X \nrightarrow N$ which represents the logical evaluation of the function $F^*(i, x) \Leftrightarrow (Fi = x)$.

The interaction between functions and relations has a mirror in the interaction between functors and profunctors. A functor $F : N \to X$ canonically generates two profunctors. One of the same direction $F_* : N \nrightarrow X$ and one of the opposite direction $F^* : X \nrightarrow N$. They are defined with the aid of hom-objects, where $F_*(x, i) = X(x, Fi)$ and $F^*(i, x) = X(Fi, x)$. In the case of Lawvere metric spaces, the profunctors of $F$ evaluated on objects $x$ and $i$ can be read as: "The distance between $x$ and the image of $i$ under $F$". With this final component, it is now possible to construct the Nearest Neighbours Algorithm.

## 3 Constructing The Nearest Neighbours Algorithm

This section explores the construction of the Nearest Neighbours Algorithm, given a dataset of points in a metric space, and classification labels, using Enriched Category Theory. Starting with a dataset of $n$ pairs $(x_1, y_1), ..., (x_n, y_n)$, the $x_i$ values are elements of a metric space $X$, and the $y_i$ values are class labels. Given a new point $x \in X$, what is the correct class label to associate with it?

From the format of the dataset, the primary characteristic of the data points is the distances between them. This would suggest that the natural choice for the enriched categories are Lawvere metric spaces, i.e. Cost enriched categories. The first step is to find an appropriate representation of the data. An individual data point, $(x_i, y_i)$, has three components. An index value $i$, an associated point in the metric space $x_i$, and the classification label $y_i$. The $n$ index values can be stored in a Cost-enriched category $N$. The metric space $X$ can clearly also be represented as a Cost-enriched category $X$, but the class labels can also be represented in a similar way, as the contents of the Cost-enriched category $Y$, which contains all of the possible class labels. With these categories, the information of the dataset can be represented by two functors. $F : N \to X$ maps the index values to their associated position in the metric space $x_i$. The functor $T : N \to Y$, similarly, maps data indexes to class labels.

Though it is now clear what objects the various enriched categories contain, it remains to determine what the hom-objects of each category should be. In the case of the metric space $X$, it is clear that between any points $a, b \in X$, the hom object $X(a, b)$ should correspond directly with the distance metric on X. It is less clear what the choice should be for the categories $N$ and $Y$.

Proceeding with the intuition that the hom-objects, or in this case the distances, between objects should encode meaningful information about the data, the objects of $N$, the indexes, possess no explicit relation to each other. This would suggest that the distances between indexes should be as "un-constraining as possible". In the context of enriched categories, the lack of constraint would suggest that the Functors from $N$ to any other Cost category, should correspond directly with maps from the objects of $N$ to the other category. To achieve this, the category $N$ can be given the discrete metric, shown in the following equation.

$$N(i, j) = \begin{cases} 0 & i = j \\ \infty & i \neq j \end{cases}$$

Recalling that functors between Cost-Categories are distance non-increasing functions, the discrete metric means that this condition is trivially satisfied, as the objects of $N$ are as distant from each other as possible. This models the lack of a relationship between the data indexes. The same logic can be applied to the objects of $Y$. Class labels should also have no meaningful relation to each other, so the discrete metric can be applied to $Y$ as well. With the categories $N$, $X$, $Y$ and the functors $F$, and $T$, the dataset can be represented by the following diagram.

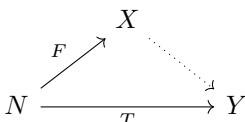

To find the classes of all the points in $X$ would optimistically be to find a suitable candidate for the dotted arrow from $X$ to $Y$. However, there is an issue. It is expected that two classification regions in $X$ may be touching, producing a boundary between classification regions which can have a trivially small distance. If we insist that classes are assigned by functors, then the functors must be distance non-increasing. This would require that the classes in $Y$ have a distance of zero from each other. It is tempting to think that one should not assign $Y$ the discrete metric, but this has an unfortunate consequence. Within the language of Enriched Category Theory, the hom-objects are the only way to distinguish between objects of a category. Setting all of the distances between objects in $Y$ to zero would make all of the classes indistinguishable from each other in any categorical construction. It was correct to assign $Y$ the discrete metric, but not to expect the classifications to be represented by a functor. The classifications can in fact be represented by a profunctor $NNA : X \nrightarrow Y$.

With the expectation that the correct classification is represented by a profunctor, we can attempt to produce this profunctor directly by composition. The functors $F$ and $T$ both have two canonical profunctors associated with them. By selecting these profunctors appropriately, we can compose them to produce a profunctor from $X$ to $Y$. This can be done with the profunctors $F^* : X \nrightarrow N$ and $T_* : N \nrightarrow Y$.

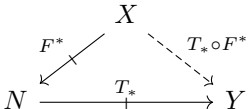

As previously discussed, the profunctor $F^* : X \nrightarrow N$ measures the distance between a point in $X$ and the image of a data point in $N$. The profunctor $T_* : N \nrightarrow Y$ does something similar, but because it is produced by a functor between discrete categories, its outputs are even easier to interpret. If a data index $i$ has a class $y$, i.e. $Ti = y$, then $T_*(y, i)$ will be 0. However, if $i$ does not have class $y$ then $T_*(y, i)$ is infinity. Substituting these profunctors into the profunctor composition formula produces the following equation.

$$(T_* \circ F^*)(y, x) = \inf_{i \in N} \left( F^*(i, x) + T_*(y, i) \right)$$

The interpretation of this composition is relatively straight forward. If the class of $i$ selected by the infimum is not $y$, then $T(y, i)$ will be infinity, making the entire sum as large or larger than any other possible value. However, if the $i$ selected was of class $y$, then the formula returns $\inf_{i \in N} F^*(i, x)$. In other words the composition $(T_* \circ F^*)(y, x)$ returns the distance from $x$ to the closest data point which is of class $y$. This could also be interpreted as evaluating the infimum of a partition of the indexes which have the class y [4].

$$(T_* \circ F^*)(y, x) = \inf_{i \in NT(y)} d(Fi, x)$$

A useful outcome, but not quite the $NNA$. There is one additional step. In order to reproduce the $NNA$ we need to compare the output of the profunctor $T_* \circ F^*$, to a similar composition with a profunctor that has no knowledge of the classes, $\mathbf{1}_{NY} : N \nrightarrow Y$.

To model the notion that $\mathbf{1}_{NY}$ has no knowledge of the classes, it must respond true to any $i \in N$ and $y \in Y$, i.e. $\mathbf{1}_{NY}(y, i) = 0$ [5]. Composing this profunctor with $F^*$ produces a composition with no knowledge of the classes.

$$(\mathbf{1}_{NY} \circ F^*)(y, x) = \inf_{i \in N} \left( F^*(i, x) + \mathbf{1}_{NY}(y, i) \right)$$
$$= \inf_{i \in N} F^*(i, x)$$

[4] Note that the following expression re-uses the notation $NT(y)$ introduced in section 2.1 to represent the partition subset of $N$ with classes $y$, $NT(y) = \{i \in N \mid Ti = y\}$

[5] This also makes $\mathbf{1}_{NY}$ the terminal profunctor of the category of profunctors between $N$ and $Y$, $Prof(N, Y)$

Given a point $x \in X$ and class $y \in Y$, the profunctor $(\mathbf{1}_{NY} \circ F^*)(y, x)$ gives the distance to the closest point in the dataset (i.e. in the image of $F$). This composition has forgotten all class information. Finally, to reconstruct the *NNA* classification it only remains to compare the outputs of both profunctors. As their outputs are objects of the Cost category, the natural comparison is their hom-object in Cost.

$$NNA : X \nrightarrow Y$$
$$NNA(y, x) := Cost((\mathbf{1}_{NY} \circ F^*)(y, x), \ (T_* \circ F^*)(y, x))$$

Because Cost can be viewed as a Bool enriched category, and therefore a preorder, this leads to the expression:

$$NNA(y, x) = (\mathbf{1}_{NY} \circ F^*)(y, x) \ \geq \ (T_* \circ F^*)(y, x)$$

A point $x$ is taken to have class $y$ when $NNA(y, x)$ is true. Consider the situation that the closest data point $Fj$ to $x$ has class $y$, then $T(y, j) = 0$. The left hand side of the inequality finds the smallest distance from x to a data point with any class and the right hand side finds the smallest distance to a data point with class y. When the closest data point to $x$ has class $y$, the left hand side returns the same value as the right hand side and the inequality is true.

$$
\begin{aligned}
NNA(y, x) &\Leftrightarrow Cost( \ (\mathbf{1}_{NY} \circ F^*)(y, x), \ (T_* \circ F^*)(y, x) \ ) \\
&\Leftrightarrow (\mathbf{1}_{NY} \circ F^*)(y, x) \quad \geq \quad (T_* \circ F^*)(y, x) \\
&\Leftrightarrow \inf_{i \in N} F^*(i, x) \quad \geq \quad \inf_{i \in N} (F^*(i, x) + T(y, i)) \\
&\Leftrightarrow F^*(j, x) \quad \geq \quad F^*(j, x) + T(y, j) \\
&\Leftrightarrow F^*(j, x) \ \geq \ F^*(j, x) \\
&\Leftrightarrow True
\end{aligned}
$$

Alternatively, in a situation where the nearest data point does not have class $y$, then $(T_* \circ F^*)(y, x) > (\mathbf{1}_{NY} \circ F^*)(y, x)$ and the output will be false. From this interpretation, it is clear that the *NNA* profunctor produces the same classification as the Nearest Neighbours Algorithm. In its purely categorical form, the similarity between the profunctor construction and the relation introduced in Section 2.1 is obscured, but it can be made clear through substitution.

$$
\begin{aligned}
NNA(y, x) &\Leftrightarrow Cost( \ (\mathbf{1}_{NY} \circ F^*)(y, x) \ , \ (T_* \circ F^*)(y, x) \ ) \\
&\Leftrightarrow (\mathbf{1}_{NY} \circ F^*)(y, x) \quad \geq \quad (T_* \circ F^*)(y, x) \\
&\Leftrightarrow \inf_{i \in N} F^*(i, x) \quad \geq \quad \inf_{i \in N} (F^*(i, x) + T_*(y, i)) \\
&\Leftrightarrow \inf_{i \in N} F^*(i, x) \quad \geq \quad \inf_{i \in NT(y)} F^*(i, x) \\
&\Leftrightarrow \inf_{i \in N} F^*(i, x) \quad = \quad \inf_{i \in NT(y)} F^*(i, x) \\
&\Leftrightarrow \inf_{i \in N} X(Fi, x) \quad = \quad \inf_{i \in NT(y)} X(Fi, x) \\
&\Leftrightarrow \inf_{i \in N} d(Fi, x) \quad = \quad \inf_{i \in NT(y)} d(Fi, x)
\end{aligned}
$$

The last line is the same as the *NNA* relation shown in Section 2.1, demonstrating that this construction is the same as the standard Nearest Neighbours Algorithm.

# 4 Generalising the Nearest Neighbours Algorithm

With the basic construction of the *NNA* presented in section 3, we now explore how this form can be generalised. Firstly, we will discuss the production of the k Nearest Neighbours Algorithm (*k-NNA*). Secondly, we will discuss forming soft or dependent classification boundaries using non-discrete metrics for classification labels.

## 4.1 K Nearest Neighbours

A common extension of the standard NNA, the $k$-NNA, bases its output on an aggregate classification formed from the classes of the $k$ nearest neighbours. To produce this generalisation, the construction needs to store the information associated with $k$ neighbouring data points, and a choice needs to be made concerning the method of aggregating the classes of these points. The first issue can be solved using the tensor product introduced in section 2.3. In particular, given a Cost enriched category $C$, we can produce the $k$ times tensor product of $C$ with itself.

$$C^k := C_1 \otimes ... \otimes C_k \qquad C = C_i$$

The tensor product operates on the objects of $C$ identically to the set-theoretic Cartesian product, meaning that the objects of $C^k$ are $k$-tuples of the objects of $C$. The tensor product also operates on hom objects of $C$ by applying the monoidal product of the base of enrichment. In the case of Cost, the monoidal product is just an addition.

$$C^k(\vec{x}, \vec{y}) = C_1(\vec{x}_1, \vec{y}_1) \otimes ... \otimes C_k(\vec{x}_k, \vec{y}_k)$$

The tensor product also acts on enriched functors similarly to how the Cartesian product acts on functions. The tensor product of functors maps tuples to tuples, with each component functor acting element-wise on the tuples. This allows the mapping produced by a functor $k$ times tensored with itself to be presented as follows.

$$F^k(\vec{x}) = (F(\vec{x}_1), ..., F(\vec{x}_k))$$

Because the tensor product of enriched categories acts on both categories and functors, we can produce the $k$ times tensor product of the diagram which defines the dataset.

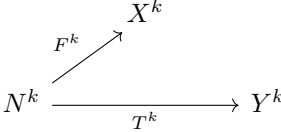

This diagram can store the information required for the $k$-*NNA* classification. However, it includes tuples that would usually be excluded, i.e., where a particular data point appears multiple times. Usually, the $k$ nearest neighbours are $k$ different neighbours. By including tuples which duplicate data points, the algorithm can base its aggregate classification on a tuple of duplicate points. To avoid this issue, we can introduce a subcategory of $N^k$, which only includes tuples with no duplicate points. This subcategory also comes equipped with a functor, which is an inclusion on objects, mapping the subcategory into the original space.

$$I : \overline{N^k} \to N^k$$

The introduction of $I$ allows the diagram to be restricted such that the algorithm only acknowledges tuples of distinct neighbours. By composing $I$ with $F$ and $T$, then taking the induced profunctors, we can produce a similar profunctor diagram as used in section 3.

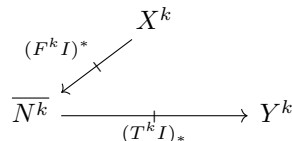

The second step in this procedure is to correct the outputs and inputs of the diagram by composing profunctors that convert $k$-tuples back into individual points of $X$ and $Y$. The $k$ times tensor product of $X$ with itself comes equipped with a profunctor functor that identifies the $k$-tuples formed from a single element of $X$.

$$\Delta : X \nrightarrow X^k$$

The profunctor identifies such tuples by comparing a single element of $X$ with each element of the tuple using the distance metric of $X$.

$$\Delta(\vec{x}, x) = X(\vec{x}_1, x) \otimes ... \otimes X(\vec{x}_k, x)$$

When every element of $\vec{x}$ is zero distance from $x$ then $\Delta(\vec{x}, x) = 0$. The profunctor used to correct the outputs of the diagram is not so clearly defined. Alternatively, it should be said that multiple viable options could be used depending on how the $k$-*NNA* is expected to operate. The aggregator profunctor takes the form $A : Y^k \nrightarrow Y$. Part of the variability in its definition is how $A$ handles ties, where the most common labels are equally prevalent amongst the $k$ neighbours. Whenever there is a clash, a greedy aggregation policy may assign both labels, a conservative policy would assign no labels, and a biased policy may always prefer one label over another. Ultimately, $A$ becomes a hyperparameter. The commonality between these schemes is that $A$ is invariant to permutations in the order of the elements, that it is 0 when a tuple of neighbours does possess the given classification label, and it is $\infty$ otherwise. The following diagram may be produced with access to the profunctors $\Delta$ and $A$.

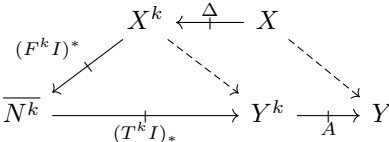

In parallel with the strategy of section 2.1 this allows the $k$-*NNA* algorithm to be defined through profunctor composition.

$$k\text{-}NNA : X \nrightarrow Y$$
$$k\text{-}NNA := A \circ Cost((\mathbf{1}_{\overline{N^k}Y^k} \circ (FI)^*), \ ((TI)_* \circ (FI)^*)) \circ \Delta$$

The definition of $k$-*NNA* can also be presented more concisely with reference to the original *NNA* algorithm, constructed over the functors $F^k I$ and $T^k I$. For clarity, we notate the original *NNA* construction as $NNA_{F,T}$.

$$k\text{-}NNA = A \circ NNA_{F^k I, T^k I} \circ \Delta$$

Presenting the $k$-*NNA* using the *NNA* construction will also allow us to leverage the arguments of section 3 without having to repeat them. Each component profunctor must yield true for the composite $k$-*NNA* to yield true. In the case of Cost-enriched profunctors, we interpret the output 0 interchangeably with the value true. To convince ourselves that the $k$-*NNA* is behaving as expected, we may follow a point $x \in X$ around the diagram.

By construction $\Delta(\vec{x}, x)$ is only true when $\vec{x}$ has as its elements objects which are zero distance from $x$. When $X$ is considered a traditional metric space, including the axiom of positivity, this forces $\vec{x}_i = x$. We also know that $NNA_{F^k I, T^k I}(\vec{y}, \vec{x})$ is true when the closest point to $\vec{x}$ in the image of $F^k I$ has the classification $\vec{y}$. All points in the image of $F^k I$ are tuples of distinct neighbours. The distance metric of $X^k$ is the sum of distances in $X$. By minimising each dimension, we may construct an object $\vec{z} \in X^k$. We select $\vec{z}_1$ to be the closest point to $x$, and then as $\vec{z}_2$ must be a different point, we select it to be the second closest point in the dataset. Iterating leads to $\vec{z}_k$ being the $k$th closest point to $x$. This shows that if $\vec{z}$ minimises the distance to $\vec{x}$, it is a tuple of $k$ nearest neighbours to $x$. The classes of $\vec{y}$ must match component-wise with each class assigned to the components of $\vec{z}$. Given a $\vec{z}$ which minimises the distance to $\vec{x}$, then a permutation of $\vec{z}$ also minimises this distance, inducing a permutation in the classes $\vec{y}$. However, this does not affect the output because $A$ is selected to be invariant to permutations. Finally, given a tuple of classes $\vec{y}$, then $A(y, \vec{y})$ is true if $y$ corresponds with the aggregate class as selected by the chosen aggregation policy. The sequence

of deductions shows that $k$-$NNA(y,x)$ yields true only when $y$ is the aggregate of the classes of $k$ nearest neighbours to $x$.

In summary, $k$-$NNA$ can be produced using the $NNA$ algorithm by duplicating the original dataset with a $k$ fold tensor product. A choice of class label aggregation policy allows $k$-tuples of labels to be reduced to a single class label. Before finishing this section, it should be noted that there are instances where a dataset may have multiple points which are equidistant from $x$. This would produce multiple tuples of neighbours, which minimise the distance without necessarily being a permutation of each other. In this case, the $k$-$NNA$ would assign multiple class labels to $x$. This can be seen as an extension of the case in the $NNA$, where a point equidistant from two data points with different classes is assigned both class labels.

## 4.2 Label Metrics for Soft Boundaries and Dependent Classifications

The categorical constructions of the $NNA$ and $k$-$NNA$ are novel to this paper, but the algorithms are already well established. While this is beneficial in justifying that $ECT$ can produce practically applicable algorithms, it would be nice to demonstrate that $ECT$ analysis can extend the utility of existing algorithms. Thankfully, the construction of the $NNA$ does not only replicate the behaviour of a traditional nearest neighbour algorithm. When the construction parameters are allowed to vary outside of what is usually possible, the $NNA$ begins exhibiting novel and potentially valuable behaviours.

The particular variation explored in this section occurs through the definition of the space of classification labels, $Y$. In the traditional implementation, the collection of classification labels is a set of values with no additional structure; as the algorithm is usually implemented directly from its description, it is unclear how one would incorporate additional structure into its decision-making. Section 3 shows that when $Y$ is considered a discrete category, the $ECT$ construction of $NNA$ reproduces precisely the expected behaviour. However, $Y$ is not required to be a discrete category in general. By framing the construction within the category of Lawvere metric spaces, the enriched $NNA$ can accept $Y$ as any Lawvere metric space. Incorporating information about the similarity of classification labels as a metric allows the enriched classifier to produce softer classification boundaries. This could be formed from a semantic similarity metric for word-based labels or a more traditional metric for vector-based labels. Metrics associated with $Y$, the category of classification labels, will be referred to as label metrics.

Label metrics are incorporated into the classification decision through $T_*$.

$$T_*(y, i) = Y(y, Ti)$$

When the label metric is discrete, $T_*$ serves only to identify when points are identical or not. However, for label metrics with bounded distances, the enriched $NNA$ may present a point as having a particular classification if that label is within a certain distance of the label of the nearest neighbour. To understand this behaviour for a non-discrete label metric, we may visualise the outputs of the $NNA$ over its entire space of input pairs. Consider, for example, a function from the unit interval to the unit interval. A sample of points from this function can be presented with functors $F$ and $T$, allowing the application of the $NNA$. Both $X$ and $Y$ values can be given the standard metric of the real numbers rather than forcing $Y$ to be discrete. The outputs of the $NNA$ and 4-$NNA$ for every $X$ and $Y$ position are visualised in Fig 2.

The images demonstrate the output values of the profunctors, showing how they evaluate the nearness to truth of each coordinate point. Operating in this fashion, the $NNA$ begins to act more as a regression model than as a classifier.

In addition to the ability to present non-discrete label metrics, the presentation of $Y$ as a Lawvere metric space allows it to adopt an asymmetric metric. This induces what one might refer to as dependent classifications, where the assignment of one label may depend on the assignment of another. For example, consider the classification labels Dog, Cat, Bird, and Mammal. The label metric between Dog, Cat, and Bird can be taken to be discrete. As all Dogs and Cats are Mammals, we can represent this relationship by allowing the metric to become zero when evaluated in one direction.

$$Y(Mammal, Dog) = Y(Mammal, Cat) = 0$$

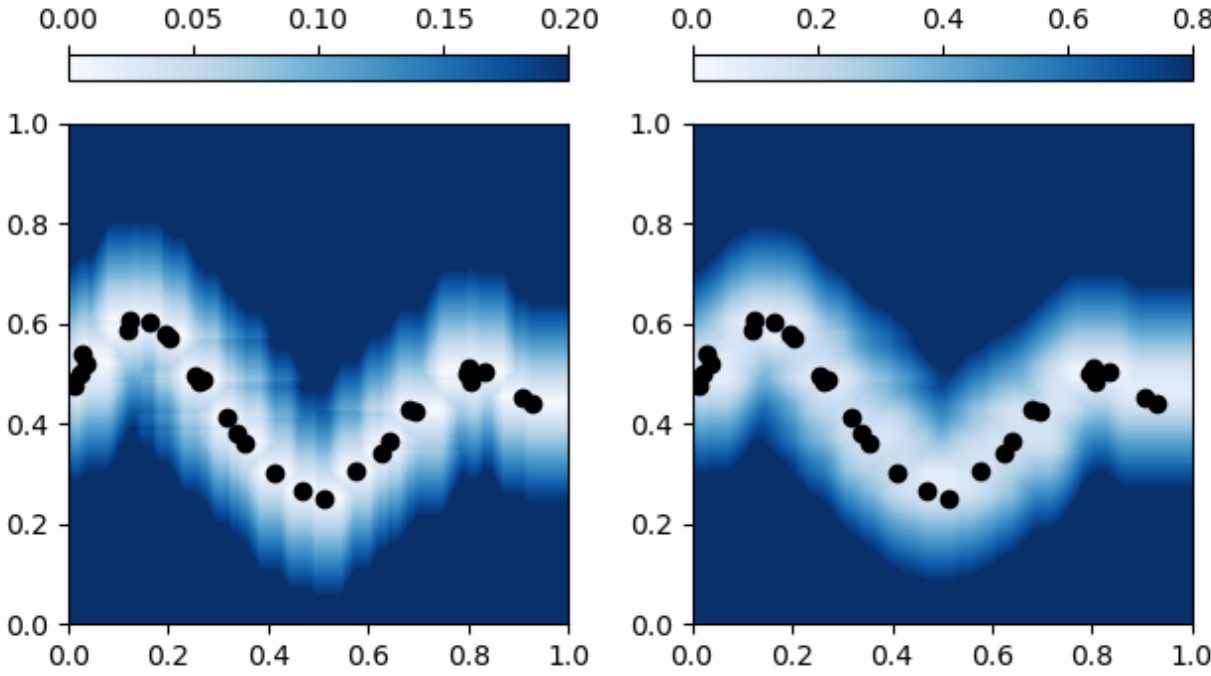

Figure 2: A plot of the values of *NNA* (left) and 4-*NNA* (right) for $x, y \in [0, 1]$. 30 points were uniformly sampled from the interval $[0, 1]$ and transformed by the function $f(x) = 0.4 + 0.1 \sin(10x) - 0.7x^2 + 0.7x^3$ then randomly scaled by $\pm 5\%$ to produce the $Y$ values. The hom objects where taken to be $X(a, b) = Y(a, b) = |b - a|$. The aggregation policy chosen for 4-*NNA* is only zero when all tuple components agree on the class. Both colour scales clip outputs outside their stated range.

As not all Mammals are Dogs or Cats, the converse relationship is assigned an infinite distance.

$$Y(Dog, Mammal) = Y(Cat, Mammal) = \infty$$

As birds are not mammals and mammals are not birds, then these labels are an infinite distance from each other.

$$Y(Bird, Mammal) = Y(Mammal, Bird) = \infty$$

Assume for some value $x$, that $NNA(x, Dog) = True$, then the two component profunctors of the *NNA* are equal.

$$(\mathbf{1}_{NY} \circ F^*)(Dog, x) = (T_* \circ F^*)(Dog, x)$$

The statement that $Y(Mammal, Dog) = 0$ allows it to be inserted into the definition of the right-hand side to demonstrate that $(\mathbf{1}_{NY} \circ F^*)(Dog, x) = (T_* \circ F^*)(Mammal, x)$.

$$
\begin{aligned}
(\mathbf{1}_{NY} \circ F^*)(Dog, x) &= (T_* \circ F^*)(Dog, x) \\
&= \inf_{i \in N} \left( F^*(i, x) + T_*(Dog, i) \right) \\
&= \inf_{i \in N} \left( F^*(i, x) + Y(Dog, Ti) \right) \\
&= \inf_{i \in N} \left( F^*(i, x) + Y(Mammal, Dog) + Y(Dog, Ti) \right) \\
&\geq \inf_{i \in N} \left( F^*(i, x) + Y(Mammal, Ti) \right) \\
&\geq (T_* \circ F^*)(Mammal, x)
\end{aligned}
$$

The presence of the infimum in the definition of $(\mathbf{1}_{NY} \circ F^*)(Dog, x)$ forces the inequality to be an equality. By the definition of the left-hand side, we know that $(\mathbf{1}_{NY} \circ F^*)(Dog, x) = (\mathbf{1}_{NY} \circ F^*)(Mammal, x)$ as $(\mathbf{1}_{NY} \circ F^*)$ is invariant to the first input. Together, these statements induce the following equality.

$$(\mathbf{1}_{NY} \circ F^*)(Mammal, x) = (T_* \circ F^*)(Mammal, x)$$

This infers that $NNA(Mammal, x) = True$. Proving the aforementioned dependent classification.

$$NNA(Dog, x) \implies NNA(Mammal, x)$$

The same logic also induces a dependent classification between Cat and Mammal. However, as the distance from Mammal to Dog or Cat is infinite, neither depends on the Mammal label. Similarly, as the labels Mammal and Bird are mutually infinite distance apart, neither depends on the other.

Exploring the enriched $NNA$ as $Y$ is allowed to take non-discrete and asymmetric metrics demonstrates that the resultant behaviours may apply to a broad range of use cases. When $Y$ is allowed to be non-discrete, the $NNA$ can make regression and classification decisions reminiscent of kernel density algorithms. When $Y$ is allowed to be asymmetric, it can encode dependent relationships reminiscent of simple ontologies. In both cases, the enrichment of the $NNA$ has unlocked behaviours with machine learning applications inaccessible to the original algorithm.

## 5 Future Work

Given the diversity of Machine Learning algorithms and the natural generalising power of Enriched Category Theory, there are numerous avenues to explore for future extensions of this work.

The construction of the $NNA$ in section 3 does not require any specific properties of Cost-enriched categories to define. This naturally leads to a candidate definition of the V-enriched Nearest Neighbours Algorithm ($V$-$NNA$).

$$V\text{-}NNA : X \nrightarrow Y$$
$$V\text{-}NNA(y, x) := V((\mathbf{1}_{NY} \circ F^*)(y, x),\ (T_* \circ F^*)(y, x))$$

The question arises as to whether this definition has valuable properties in other bases of enrichment. Though the previous section interpreted the hom-object of the base of enrichment in its Bool-enriched form for the sake of clarity, future works would benefit from considering the self-enriched form of the hom-object and, in the case of the Cost-NNA, interpreting the hom-object with truncated subtraction rather than an inequality.

$$Cost\text{-}NNA : X \nrightarrow Y$$
$$Cost\text{-}NNA(y, x) = (T_* \circ F^*)(y, x) \mathbin{\dot{-}} (\mathbf{1}_{NY} \circ F^*)(y, x)$$

Researchers who are not interested in Machine Learning would possibly consider the Voronoi diagram as a more exciting outcome of the $V$-$NNA$. By assigning each index a separate class, the $NNA$ partitions the metric space dependent on each point. In this instance, the partitions generated in other bases of enrichment may prove interesting.

The generalisation of the $NNA$ to the $k$-$NNA$ shows how the latter can be expressed in terms of the former using the flexible language of profunctor composition. This relationship is not clear in other presentations. Further investigation of how profunctor composition can be used to express machine learning algorithms may lead to deeper insights. Beyond this, the increased flexibility of the enriched $NNA$ in allowing $Y$ to take on a variety of metrics requires practical verification to demonstrate its utility on real-world datasets.

## 6 Conclusion

The nascent field of Category Theory for Machine Learning has grown in recent years. As Category Theory is predominantly concerned with mathematical structure, there is hope that such techniques can improve our understanding of Machine Learning algorithms. Previous works have demonstrated that there is value in this avenue of research. However, there are currently not enough examples to indicate the correct way to apply Category Theory to understanding Machine Learning algorithms. In particular, there has not previously been an application of Enriched Category Theory in Machine Learning. With the construction of the Nearest Neighbours Algorithm, using tools from Enriched Category Theory, there is now a stronger indication that this area can provide valuable insight. Furthermore, the strategies used to represent information and to reason about the construction of machine learning algorithms in this format suggest that the enriched structure offers a potentially more intuitive framework than other categorical attempts.

The simplicity of constructing the Nearest Neighbours Algorithm in this framework adds credence to the sense that the algorithm is an exceedingly natural approach to extending classifications. With the formulation of the Enriched Nearest Neighbours Algorithm, it becomes a tantalising area of future work to ask if this algorithm continues to provide sensible classifications in other bases of enrichment. This motivation is part of the underpinning interest mentioned in the introduction of this work. Is it the case that machine learning requires fundamentally new algorithms to tackle stranger and stranger problems? Alternatively, when suitably abstracted, a handful of algorithms might be sufficient for most cases, and the engineering challenge comes in choosing the correct base of enrichment.

Another interesting outcome of this work is to indicate that Enriched Category Theory is a framework of reasoning that should be of more interest to Machine Learning researchers. Often derided as a more abstract formulation of the exceedingly abstract field of Category Theory, certain enrichment bases create enriched categories that are practically useful. Furthermore, it indicates that understanding the interaction between hom-objects, functors, and profunctors can provide valuable insights into structuring information and the meaning behind those structures. Even if one does not find the rigorous application of the theory useful, the intuition may prove helpful.

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
