# OpenReview forum: "Using Enriched Category Theory to Construct the Nearest Neighbour Classification Algorithm"
_TMLR — Rejected by TMLR_

### Review · Reviewer_LZE7 · 2024-09-22

**Summary Of Contributions:**

The paper presents a new way to construct the popular nearest neighbor algorithm (NNA) in machine learning using Enriched Category Theory (ECT) in the category theory. The key idea is to formalize NNA on a Lawvere metric space, which allows categories to operate like metric spaces, with functors $F:N \to X$ and $T:N \to Y$, and this expression is given in page 7. The authors briefly mention potentially interesting areas that can leverage ECT.

**Audience:**

No

**Broader Impact Concerns:**

This paper is a purely theoretical work. I do not find any specific ethical or social concerns related to this work.

**Claims And Evidence:**

Yes

**Requested Changes:**

I did not find any significant mathematical errors in this paper. However, at the same time, I did not find any interesting results in this paper that were worthy of publication in TMLR. This paper needs to be significantly revised to make it a high-quality work and solid contributions to ML community. With the current form, it remains unclear what theoretical or empirical insights we can further achieve from this construction. Please see the Weaknesses section together.

- (minor) Given that most TMLR readers are likely unfamiliar with category theory, a table comparing the notations of set theory (e.g., map) and category theory (e.g., functor) would be very helpful.

**Strengths And Weaknesses:**

I should first acknowledge that I am not familiar with a category theory, but it is not very difficult to understand the mathematical findings in this paper.

##  Strengths
- The paper provides a new way to interpret the NNA algorithm from a category theory perspective.

## Weaknesses
- (major) The current construction seems to be purely mathematical, which is not necessarily bad, but it is not clear exactly what further insights this construction may offer.
  - Related to this point, the current construction is too specific to NNA. As alluded by authors in Section 4, generalization to the $k$-nearest neighbor algorithm with $k>1$ would provide more interesting findings.
  - Extension to other machine learning algorithms is not clear.

- Undefined notations and typos (This is supposed to be a minor point, but there are too many typos. This is definitely not something you would expect to see in a high-quality research paper.)
  - Throughout the paper, abbreviations (e.g., NNA, ECT) were sometimes not defined when first mentioned, or were not used after being defined in a previous section.
  - interpretted (6 times) --> interpreted
  - Inconsistent use: Neighbours (21 times) vs Neighbors (1 time)
  - Section 1: "How does the encoding of input data **effect** the information" --> "How does the encoding of input data **affect** the information"
  - Section 2.3: $Y^{op}$ is not defined.
  - Section 3: The first paragraph of Section 3: $n$ pairs --> $n+1$ pairs (as it starts from 0).
  - Section 3: "the primary characteristic of the data points **are** the distances between them" --> "the primary characteristic of the data points **is** the distances between them"

- (minor) Figure 1 does not look necessary or critical. Also, the experimental setting is not clearly provided (e.g., mean, variance, the number of samples, and the proportion of positive class).

---

> ### Author Response · Authors · 2024-10-02
> **First response to reviewer LZE7**
>
> Thank you for reviewing my paper. I have fixed the typos and notation issues in the newly submitted revision and noted how I have addressed each of your comments in a seperate response.
>
> I completely understand where you are coming from regarding the more significant concerns you have raised. I also noticed in your review that you identified the specificity of the construction as an issue because the results would not be suitably interesting to the TMLR audience. However, I think that viewing this paper in the context of the broader literature will show that its contents cater to a significant gap.
>
> A growing community of category theorists are producing papers analysing machine learning algorithms with categorical methods. They are attempting to respond to a call from the machine learning community for better ways to understand the behaviour of machine learning algorithms, given their structure and the structure of the datasets they operate on. However, almost all these papers are abstract and rely considerably on categorical concepts. They are inaccessible to a large portion of the audience that might have a use for their insights and are also largely uninteresting to this community, as their results don't easily translate to practical applications. This is the hole that the paper is attempting to begin covering. Offering a textbook example that can provide an introductory stepping stone to the concepts and their applications. After reading it, there will hopefully be some portion of the readership that will: firstly, know of at least one example where these techniques produce an algorithm they are confident can be practically applied; secondly, be aware of some categorical concepts alongside an incentive to engage further with the research as they spot applications to their interests.
>
> In terms of generalising this result to other algorithms, I think this becomes clearer by observing the general pattern the paper describes. One of the uses of enriched category theory is that it characterises predicates and relations that possess alternate truth values. Notionally, if we are attempting to produce machine learning algorithms which make decisions based on available information, we are designing them to emulate logic circuits, though not necessarily using Boolean logic. If you refer to the construction of a profunctor and its composition, you'll notice that not only does it generalise a normal set relation, but it also appears to include the description of matrix multiplication with a non-linear activation function (à la neural networks). This indicates that the design of a neural network reflects how it encodes logical formulae of alternate logic systems as a profunctor composition. Future work is needed to confirm this relationship. By introducing these patterns to a broader machine-learning community, we may be able to uncover which aspects of machine-learning design are patterns of alternately valued logic.
>
> Considering your comments, it appears that the paper does not currently communicate its value and how it fits within the broader literature. I'm hopeful that including a few paragraphs with this perspective in mind may better convince future readers that its contents can be of use to them and, by extension, may convince you that it will be of interest to the TMLR readership.

---

> ### Author Response · Authors · 2024-10-02
> **Revision modifications in response to reviewer comments**
>
> 1. Undefined notations and typos (This is supposed to be a minor point, but there are too many typos. This is definitely not something you would expect to see in a high-quality research paper.)
>
> - I believe I have now fixed all the notational issues and typos that you have mentioned
>
> 2. Throughout the paper, abbreviations (e.g., NNA, ECT) were sometimes not defined when first mentioned, or were not used after being defined in a previous section.
>
> - I have added the definition for ECT to the introduction. The definition for NNA already appears in the introduction, but I can add the definitions to subsequent sections if you think this would improve the article.
>
> 3. interpretted (6 times) --> interpreted
>
> - Changed all instances of interpretted to interpreted.
>
> 4. Inconsistent use: Neighbours (21 times) vs Neighbors (1 time)
>
> - Changed Neighbors to Neighbours for consistency.
>
> 5. Section 1: "How does the encoding of input data effect the information" --> "How does the encoding of input data affect the information"
>
> - Changed effect to affect.
>
> 6. Section 2.3: Y^op is not defined.
>
> - I have added an explanation of Y^op to section 2.3 where it is first used. Y^op is a category with the same objects as Y, but the hom objects are reversed, so Y(a, b) = Y^op(a, b).
>
> 7. Section 3: The first paragraph of Section 3: pairs --> n+1pairs (as it starts from 0).
>
> - I have changed the pairs to be 1 indexed rather than 0 indexed to fix this issue.
>
> 8. Section 3: "the primary characteristic of the data points are the distances between them" --> "the primary characteristic of the data points is the distances between them"
>
> - Changed “are” to “is”
>
> 9. (minor) Figure 1 does not look necessary or critical. Also, the experimental setting is not clearly provided (e.g., mean, variance, the number of samples, and the proportion of positive class).
>
> - You are right, it is not necessary or critical. I included Figure 1 with the intention of improving the readers experience by providing a general example of what a nearest neighbours classification may look like. This is also why I didn’t include the experimental setting. If you feel that the figure should not be included or that the experimental setting is required then I can make these modifications.
>
> 10. (minor) Given that most TMLR readers are likely unfamiliar with category theory, a table comparing the notations of set theory (e.g., map) and category theory (e.g., functor) would be very helpful
>
> - I have added the table at the end of section 2.

---

> > ### Comment · Reviewer_LZE7 · 2024-10-04
> > **Response to Official Comment**
> >
> > I appreciated the authors' detailed responses. The authors addressed many of my minor points, and I agree with the authors that this paper could be of interest to some audience in the broader context. However, the current responses did not clearly answer my major comment "it is not clear exactly what further insights this construction may offer."

---

> > > ### Author Response · Authors · 2024-10-04
> > > **Second response to reviewer LZE7**
> > >
> > > Thank you for allowing me to cover this gap in my response. There is one clear insight that can be taken from this construction, which, in hindsight, should be discussed more clearly in the paper.
> > >
> > > By framing the construction in the category of Lawvere metric spaces, not only do the x values occupy a metric space, but the classification labels do as well. The construction takes the metric space of classification labels as discrete to reproduce the original nearest neighbours classifier identically. However,  the space of classification labels does not have to be discrete. This allows the enriched classifier to produce softer classification boundaries, given prior knowledge of the similarity between classification labels. This could be formed from a semantic similarity metric for word-based labels or a more traditional metric for vector-based labels.
> > >
> > > Additionally, the label similarity metric does not need to be symmetric. This allows for hierarchical classification labels. An example may be an ontology where some labels imply others. Given a label 'dog' and a label for 'animal', the distance from 'dog' to 'animal' may be zero, as dogs are animals. However, the distance from animals to dogs will be much larger as many animals are not dogs. This would lead the enriched NNA always to classify a point as an animal if it has classified it as a dog, but not necessarily the other way around.
> > >
> > > If you find this idea important in justifying the value of the enriched NNA, I can add paragraphs or a new section exploring this feature.

---

### Review · Reviewer_BUkR · 2024-09-23

**Summary Of Contributions:**

This paper explores the use of Enriched Category Theory (ECT) as a foundation for developing and analyzing machine learning algorithms. Specifically, it demonstrates how the Nearest Neighbors Algorithm (NNA) can be constructed using ECT concepts. The authors argue that using ECT can lead to more transparent and interpretable machine learning algorithms, e.g., by making assumptions about datasets more explicit. The paper begins with an introduction to ECT, focusing on Lawvere metric spaces as Cost Enriched Categories. It then proceeds to construct the NNA step-by-step using ECT concepts such as functors, profunctors and their compositions. The authors show that their ECT-based construction yields the same classification as the standard NNA. The paper concludes by discussing potential future work, including generalizing the NNA to other bases of enrichment and outlining the application of ECT to other machine learning algorithms.

**Audience:**

Yes

**Claims And Evidence:**

No

**Requested Changes:**

It seems that some claims of the paper on the benefits and potential of this approach are not sufficiently supported, for instance:

1) More explanations are needed to demonstrate the promised benefits of this approach, i.e., clear evidence how the ECT contruction of NNA leads to more "robust" and "explainable" algorithms, that "draw a tighter parallel between the intuition of how the dataset is represented".

2) More examples of algorithms are needed to support the claim that ECT might serve as a "foundation of an alternative approach to Machine Learning".


**Minor issues and questions:**

- Two sentences in the abstract seem to be incomplete: "Exploring whether Enriched Category Theory could provide the foundation of an alternative approach to Machine Learning." and "In particular, as an extension of the original dataset using profunctors in the category of Lawvere metric spaces."
- Typo: interpretted
- Use `\citep` for parentheses around the citation.
- $n$ should be used instead of n for the number of pairs.
- In the current formulation, NNA is defined as a relation that is potentially true for two classes if the point lies at the decision boundary; however, this makes it problematic to say that NNA "*assigns* a class if the closest point in the dataset has that class." or "A point $x$ is taken to have class $y$".
- Do not mix first person singular "I" and plural "we" in the presentation.
- It should be better explained what is meant by arrows of categories, e.g., in "Bool also has arrows of implication..." or "the arrows of the Cost category".
- Why can we interpret $Cost(a, b) = a \ge b$, i.e., having a Boolean output instead a non-negative real number?

**Strengths And Weaknesses:**

**Strengths:**

- Novelty: The paper presents one of the first examples of constructing a machine learning algorithm solely using Enriched Category Theory, which is a novel and potentially valuable approach.

- Accessibility: The authors provide an intuitive step-by-step explanation of the NNA construction using ECT, making it accessible to readers with limited background in category theory.

- Potential for generalization: The paper suggests a general form for the V-enriched Nearest Neighbours Algorithm, which could lead to interesting variations of the algorithm in different bases of enrichment. Moreover, the paper draws connections to Voronoi diagrams and outlines avenues for future work.

**Weaknesses:**

- Limited implications: While the paper demonstrates the theoretical construction of the NNA using ECT, it does not discuss any new insights or practical advantages obtained from this construction.
Although the authors claim that ECT can lead to more interpretable algorithms, they do not provide a detailed discussion of how their construction improves interpretability. Similarly, while robustness of algorithms is mentioned as a motivation in the abstract, this is not discussed in the main paper.

- Narrow focus: The paper focuses solely on the NNA, which is one of the simplest machine learning algorithms. It remains unclear if and how this approach would be applied to more complex algorithms.

- Limited discussion of related work: While related work is mentioned, more details on the similarities and differences to previous work and other categorical approaches (e.g., using operads or monoidal categories) should be provided.

---

> ### Author Response · Authors · 2024-10-02
> **First Response to Reviewer BUkR**
>
> Thank you for reviewing my paper. I have submitted a revised paper and noted how I have addressed each of your comments in a seperate response.
>
> Regarding your more significant concerns, I understand why you would say that the paper's contents may not adequately support this portion of the claims. Part of the goal of this paper was to serve as an introduction to an enriched category theory approach to analysing machine learning algorithms. The claims you have identified were partly intended as an aspirational claim for the future potential of ECT within the context of machine learning. In particular, the claim for robustness is not obviously supported by the paper's contents. However, I think that the other two claims of explainability and to "draw a tighter parallel between the intuition of how the dataset is represented" can be supported by the results presented.
>
> Firstly, as a case for the claim of explainability, I would argue that the construction of the NNA allows the operating principle of the classifier to be more directly phrased as a human-interpretable logical statement. Enriched Category Theory assists in characterising predicates in alternate logic systems, with the Cost enriched profunctor taking the role of a binary predicate whose outputs can be interpreted as distances from true. The first predicate of the construction (1_{N Y} ◦ F ∗)(y, x) evaluates whether the value x has already been observed. Its logical truth value is consequently the distance of x to the nearest point in the dataset. The second predicate (T_∗ ◦ F^∗)(y, x) evaluates whether x has been observed with class y. The hom object of Cost is treated as the Lawvere metric space version of logical implication. This allows the complete formula Cost( (1N Y ◦ F^∗)(y, x) , (T_∗ ◦ F^ ∗)(y, x) ) to be the evaluation of the sentence "the closest value in the dataset to x has the same class as y". Granted, it's not the greatest example of explainability in machine learning, but it does provide a bridge between the human expression of a statement and the computation of its answer.
>
> Secondly, regarding the claim "draw a tighter parallel between the intuition of how the dataset is represented", I believe this is done by the formulation of the dataset by the functors F and T. The information included in the dataset is points in a metric space with classification labels. The question that the algorithm attempts to answer is how the positions of the points infer their classification. How does x in X imply y in Y? Solving the extension problem induced by F and T directly answers the question, "How are the dependent variables of the dataset affected by the independent variables?" The sequence of steps which transition between having such a dataset, representing it with F and T, seeing that the question of how T depends on F is an extension problem, and then solving that extension problem is, to me, a very natural sequence. The procedure for transitioning from the dataset to the answer is a natural progression without jumps in reasoning.
>
> I can see how our perspectives on how this evidence services the claims might differ. I am very happy to adjust the claims or include additional expository sections depending on your preference.

---

> ### Author Response · Authors · 2024-10-02
> **Revision modifications in response to reviewer comments**
>
> 1. Two sentences in the abstract seem to be incomplete: "Exploring whether Enriched Category Theory could provide the foundation of an alternative approach to Machine Learning." and "In particular, as an extension of the original dataset using profunctors in the category of Lawvere metric spaces."
>
> - I’m not sure what the issue with the sentences is, but I’m happy to change them to whatever you’d prefer.
>
> 2. Typo: interpreted
>
> - Changed all instances of interpretted to interpreted.
>
> 3. Use \citep for parentheses around the citation.
>
> - Changed \cite to \citep
>
> 4. n should be used instead of n for the number of pairs.
>
> - Changed n to $n$ in section 2.1
>
> 5. In the current formulation, NNA is defined as a relation that is potentially true for two classes if the point lies at the decision boundary; however, this makes it problematic to say that NNA "assigns a class if the closest point in the dataset has that class." or "A point is taken to have class".
>
> - This is a good point. Currently, if there are multiple closest points with different classes the NNA will assign both classes. So a point may possess multiple classes (which is a desirable notion for boundary cases). I have changed the first sentence to "assigns a class if a closest point in the dataset has that class." I think that technically it is still correct to say "A point is taken to have class" as it does not infer that the point has uniquely that class. Please advise if you have a preferred way to fix this issue.
>
> 6. Do not mix first person singular "I" and plural "we" in the presentation.
>
> - Changed all instances of “I” to “we”
>
> 7. It should be better explained what is meant by arrows of categories, e.g., in "Bool also has arrows of implication..." or "the arrows of the Cost category".
>
> - This is my mistake in trying to present only the necessary components of the theory to understand the paper. Bases of enrichment for enriched category theory come from ordinary categories with certain properties (such as the monoidal product). This gives every base an underlying category whose structure is presented using arrows, also called morphisms, between objects. These arrows can be composed which produces the combinatorial structure that defines a category. When I refer to the arrows of Bool, or that Cost can be interpreted as having a Boolean output (as mentioned below) I am referring to the existence of a unique arrow (in these cases) which serves as a witness to the logical value of the statement. To fix this, I have removed references to the underlying category and its arrows in the text. Instead, I have shown that Cost can be described as a Bool enriched category, which builds on notions already introduced to the reader.
>
> 8. Why can we interpret , i.e., having a Boolean output instead a non-negative real number?
>
> - As described above, Cost can be represented as a Bool enriched category, allowing its hom objects to take Boolean values.

---

### Review · Reviewer_DS1Z · 2024-10-18

**Summary Of Contributions:**

The authors introduce a generalization of nearest neighbour based algorithms which uses enriched category theory. The definition uses a potentially rich structure of the label space Y.

**Audience:**

No

**Broader Impact Concerns:**

I don't see any broader impact concerns.

**Claims And Evidence:**

No

**Requested Changes:**

At this point I don't think that the paper makes a good enough case that the work is interesting to a Machine Learning audience.

This work needs a proper related work section in which the authors compare their work to existing approaches in the literature.

presentation and positioning:
-please properly formulate the problem in machine learning you are trying to solve. What is the input? What is the output?
-What are other approaches for solving this problem? Your work currently only generalizes nearest neighbour and afterwards gives examples for generalization. You then go to label spaces with a rich structure as an example of potential use case. Is this a learning problem that already exists in the literature? If so, what are the challenges and how is your algorithm addressing them?
- How would a more naive approach fail and what does enriched category present that is new. If the learning problem you are presenting is indeed novel, argue that it captures interesting problems in reality and make it a central part of your contributions.

more analysis needed:
 What are obstacles to generalization? When is generalization possible?

Is the structure in the label space assumed to be known? Is this well-motivated in practice to have full knowledge of such a rich structure? If not, can the structure be inferred?

Argue that enriched category theory is indeed a reasonable formalization and not unnecessarily complicated.
For example, it is already common to formulate machine learning problems in terms of a pointwise general loss, which takes as input an classification rule/hypothesis h, an instance x\in X and a label y\in Y and outputs a real number that measures whether h makes a good prediction for the  pair (x,y). The goal of ML algorithms is usually to have high expectation low expected loss over some "ground truth" distribution over X\times Y. While losses are often assumed to have a simple structure, they are not necessarily symmetric or fulfill any other properties of a metric. How does category theory expand over doing more common analysis with less common/more general losses and outputs (i.e. sets of labels rather than labels)?

My guess is that in this paper you are doing multi-label classification, with potentially non-standard loss functions. There is already work on nearest neighbour algorithms for multi-label classification (https://www.sciencedirect.com/science/article/abs/pii/S0031320307000027?via%3Dihub). You need to do extensive literature search and comparison to check what loss functions have been analysed for multi-label prediction.

**Strengths And Weaknesses:**

Strengths:

The application of enriched category theory to machine learning is novel.

There don't seem to be any technical errors in the paper.

Their algorithms can adapt to structured label spaces.

Weaknesses:
presentation and positioning: The paper is not well-positioned within the Machine learning literature. There is no related work section. The claims in the intro about using machine learning as a black box, is true for some applications, but not for others which are based on prior knowledge. There is a significant amount of literature that tries to capture the prior knowledge needed to generalize.
The paper in general does not seem to be written for a machine learning audience, but rather for people who want to find applications of enriched category theory. There should be a greater emphasis on what problems this approach is trying to solve and how current applications fail to do so.

A Machine learning algorithm is only successful if it successfully generalizes from finite data to a general rule. This work does not provide any evidence that their algorithms would be capable of providing data-efficient generalization. The emphasis of the paper is only on defining the algorithm and argue about the expressibility.

I'm not yet convinced of the usefulness of the generalized algorithm. Most of what the paper does is redefine existing algorithms/argue that  the standard nearest neighbour algorithm can be recovered. In the revision the authors do give an example which goes beyond standard nearest neighbour, which requires the label space to be highly structured. However this is presented more as an afterthought and it is unclear to me whether the learning problem which assumes such a rich structure is plausible.

The ML problem to be solved is under defined. The main use-case that is argued beyond re-creation on NN and k-NN is one where the label space Y is not binary and has a richer structure, i.e., labels are hierarchical with labels such as "dog", "cat", "mammal", "bird". In the example it looks like an instance can have two labels at once (e.g. "dog" and "mammal" at the same time). This would then constitute a different learning problem than if the goal was to simply predict one label and should be properly formalized.

---

### Decision · Action_Editor_spr5 · 2024-11-21

**Recommendation:** Reject

**Comment:**

The potential use of enriched category theory might be interesting to machine learning. However, this submission does not make a convincing case for this, as detailed above. The authors may consider submitting a major revision. This revision would need to include clear and substantiated evidence that the use of enriched category theory to describe ML algorithms can lead to new insights or understanding. In addition, the scope of the work should be broader than a single algorithm.

**Audience:**

The scope of the paper is too narrow to be of interest to the ML community.

**Claims And Evidence:**

The authors claim that enriched category theory can be used to provide new understanding of ML algorithms. However, the paper does not support the claim, as it is focused on translating one classical algorithms (k-NN) to the language of category theory, without providing convincing evidence that this leads to any new insights. In addition, relevant related works which would put this work in context are not discussed.

**Resubmission Of Major Revision:**

The authors may consider submitting a major revision at a later time.